# Low genetic variation is associated with low mutation rate in the giant duckweed

Shuqing Xu [1], Jessica Stapley [2], Saskia Gablenz[3], Justin Boyer[3], Klaus J. Appenroth[4], K. Sowjanya Sree[5], Jonathan Gershenzon[3], Alex Widmer [6] & Meret Huber [3,7]

Mutation rate and effective population size ($N_e$) jointly determine intraspecific genetic diversity, but the role of mutation rate is often ignored. Here we investigate genetic diversity, spontaneous mutation rate and $N_e$ in the giant duckweed (*Spirodela polyrhiza*). Despite its large census population size, whole-genome sequencing of 68 globally sampled individuals reveals extremely low intraspecific genetic diversity. Assessed under natural conditions, the genome-wide spontaneous mutation rate is at least seven times lower than estimates made for other multicellular eukaryotes, whereas $N_e$ is large. These results demonstrate that low genetic diversity can be associated with large-$N_e$ species, where selection can reduce mutation rates to very low levels. This study also highlights that accurate estimates of mutation rate can help to explain seemingly unexpected patterns of genome-wide variation.

[1] Institute for Evolution and Biodiversity, University of Münster, Hüfferstrasse 1, 48149 Münster, Germany. [2] Center for Adaptation to a Changing Environment, ETH Zurich, Universitätstrasse 16, 8092 Zürich, Switzerland. [3] Department of Biochemistry, Max Planck Institute for Chemical Ecology, Hans-Knöll-Strasse 8, 07745 Jena, Germany. [4] Matthias-Schleiden-Institute, Plant Physiology, Friedrich Schiller University of Jena, Dornburgerstraße 159, 07743 Jena, Germany. [5] Department of Environmental Science, Central University of Kerala, Periye 671316, India. [6] Institute of Integrative Biology, ETH Zurich, Universitätstrasse 16, 8092 Zürich, Switzerland. [7] Institute of Plant Biology and Biotechnology, University of Münster, Schlossplatz 7, 48143 Münster, Germany. Correspondence and requests for materials should be addressed to S.X. (email: shuqing.xu@uni-muenster.de) or to M.H. (email: huberm@uni-muenster.de)

Explaining within-species genetic diversity—measured as the level of intraspecific DNA sequence variation—is one of the major goals in evolutionary and conservation biology[1,2]. While intraspecific genetic diversity is known to vary widely among species, the underlying causes remain controversial[3,4]. According to population genetic theory, the population mutation parameter ($\theta$) is determined by the product of the spontaneous neutral mutation rate ($\mu$) and effective population size ($N_e$), and in diploid species $\theta = 4 \times N_e \times \mu$[5]. In practice, the parameter $\theta$ is often estimated by the average pairwise nucleotide diversity ($\pi$) at putatively neutral sites[6]. While the role of $N_e$ in explaining variation in genetic diversity among taxa has received much theoretical and empirical attention, the influence of variation in mutation rate remains largely ignored[3,4,7].

As most spontaneous mutations are deleterious, selection should favor lower mutation rates, but in small populations the efficacy of selection to lower the mutation rate is limited ($s \ll 1/N_e$, where $s$ is the selection coefficient against the increase of mutation rate) as genetic drift overrides the effect of natural selection. This 'drift-barrier' hypothesis can explain variation in mutation rates and the observed logarithmic-scaled negative relationship between $N_e$ and $\mu$ among species[8]. An important prediction of this model is that a large effective population size could result in the evolution of a low mutation rate. One consequence of this is that populations with very large effective population sizes may have very low genetic diversity, when selection has driven mutation rate to an extremely low level. However, to our knowledge, whether this pattern is present in eukaryotes is unknown, largely due to the paucity of studies quantifying both genome-wide diversity and spontaneous mutation rates.

To better understand the relationship between genetic diversity, mutation rate, and $N_e$, we independently obtained genome-wide and range-wide estimates of genetic diversity and mutation rate in the diploid freshwater plant *Spirodela polyrhiza* L. (Schleid.) ("giant duckweed"). This species is one of the fastest growing angiosperms; under suitable growth conditions, it reproduces predominantly by asexual budding with a duplication rate of 2–3 days[9,10]. Consequently, *S. polyrhiza* often achieves extremely high census population sizes in nature as millions of individuals can be found in a single pond. However, previous studies using a limited number of genetic markers found low genetic diversity[11,12], and whole genome resequencing of two genotypes revealed overall low heterozygosity[13]. Here, by sequencing 68 world-wide distributed genotypes and measuring spontaneous mutation rate under natural conditions, we show that low genetic variation in *S. polyrhiza* is associated with low mutation rate in the giant duckweed.

## Results and Discussion

### Genetic diversity in *S. polyrhiza*.
To provide genome-wide and range-wide estimates of genetic diversity in *S. polyrhiza*, we sequenced the genomes of 68 genotypes representing the global distribution of the species, using Illumina short-read sequencing with 29× average coverage (Supplementary Data 1). All sequence reads were aligned to the *S. polyrhiza* reference genome[14] using the BWA-MEM aligner and genetic variants were identified using GATK[15]. In total, we found 996,115 biallelic and 7,880 multiallelic high-quality single nucleotide polymorphisms (SNPs) as well as 214,262 small indels. This represents on average one SNP per 145 bp in the *S. polyrhiza* genome, which is low compared to an average of one SNP per 23 bp in *Arabidopsis thaliana* when a comparable number of genotypes are sequenced[16]. Among all biallelic SNPs, 14,191 nonsynonymous and 8865 synonymous SNPs were found (Supplementary Table 1 and Supplementary Data 2). The estimated

*S. polyrhiza* range-wide pairwise nucleotide diversity at synonymous sites ($\pi_S$) was 0.00093, which is among the lowest values reported for any multicellular eukaryote for which genome-wide genetic diversity has been estimated (see Supplementary Data 3)[3].

Population structure analysis based on genome-wide polymorphisms revealed four population clusters in *S. polyrhiza*, which are centered in four geographic regions: America, Europe, India, and Southeast (SE) Asia (Fig. 1). A few samples showed discrepancies between their geographic origin and population cluster assignment based on their genomic variation, likely due to either recent migrations of the duckweed associated with human activities or mis-labeling during long-term maintenance of the duckweed collections. The pairwise $F_{st}$, an indicator of relative differentiation between populations, ranged from 0.35 to 0.82 (Supplementary Table 2), suggesting distinct regional populations in *S. polyrhiza*. Between populations, the genome-wide nucleotide diversity from all sites ranged between 0.00067 (SE Asian versus European population) and 0.00013 (European versus American population). Within populations, $\pi$ calculated from all sites ranged from 0.00018 (American population) to 0.00056 (SE Asian population) (Fig. 1b, Supplementary Table 3). The extent of linkage disequilibrium (LD) declined with physical distance between linked loci and this rate of decline varied between populations (Fig. 1c). The average distance between SNPs with an LD coefficient ($r^2$) of 0.33 varied from 8.6 kb in the SE Asian population to 86.8 kb in the European population. The relatively slow decay of LD in *S. polyrhiza* may be attributed to its predominantly clonal reproduction. Comparing across populations, we observed much faster LD decay in the SE Asian population, suggesting more frequent (historical or ongoing) sexual reproduction in this region and/or higher $N_e$. Together, these results establish that genome-wide nucleotide diversity in *S. polyrhiza* is extremely low and sexual reproduction might be frequent in the SE Asian population.

### Mutation rate and effective population size in *S. polyrhiza*.
To investigate if the observed low genomic diversity in *S. polyrhiza* can be explained by universally low mutation rate or, alternatively, low effective population size, we estimated the spontaneous mutation rate and used our estimates of mutation rate and genomic diversity to estimate effective population size. Because environmental factors, such as ultraviolet (UV) light, which prevails in the native habitats of *S. polyrhiza*, can affect mutation rates[17–21], mutation rates measured in the lab may not necessarily reflect mutation rates in nature. In an attempt to get an estimate of mutation rate more similar to what would be observed in nature we estimated the genomic mutation rate in indoor and outdoor mutation accumulation (MA) experiments, and manipulated UV light in the outdoor experiments to further assess the effect of environmental factors (see Supplementary Fig. 1 and Supplementary Data 4). Offspring of a single common ancestor were propagated as single descendants under these conditions for 20 generations (see Supplementary Fig. 2), after which individual plants from five replicates per treatment were collected, and their genomes sequenced and compared to the ancestral genome. We obtained genome information for 16 individuals (including the common ancestor) with an average coverage of 28× (Supplementary Table 4) and identified genetic variants in more than 79.7% of the *S. polyrhiza* genome (~126 Mb). Among the 15 offspring, four de novo mutations were identified and confirmed by Sanger sequencing. These mutations all originated from the outdoor MA experiments, and located in non-coding regions. One mutation (C:G→T:A) was found in a UV-shielded line and the other three mutations (two C:G→T:A and one C:G→A:T) were found in UV-exposed lines (Table 1).

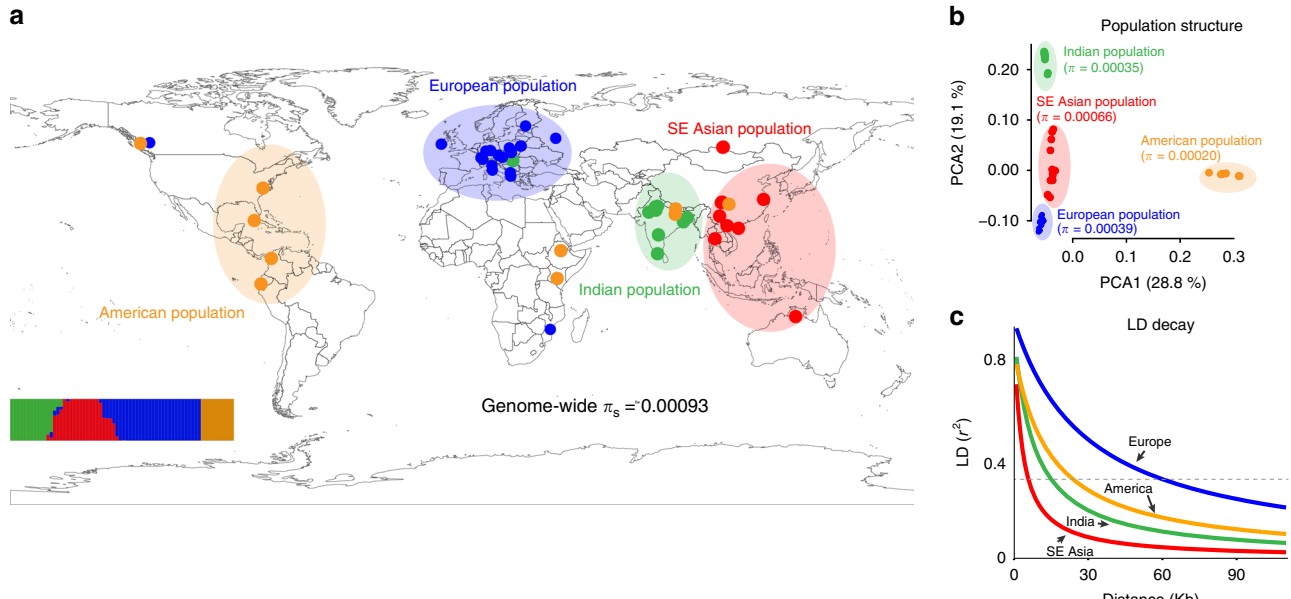

**Fig. 1** Nucleotide diversity, population structure and linkage disequilibrium in *S. polyrhiza*. **a** Geographic distribution of the 68 sequenced samples, colored according to population structure. The insert at the lower left corner shows the results from the STRUCTURE analysis using genome-wide polymorphisms. Each colored line refers to an individual and the *Y*-axis refers to the likelihood of membership to each cluster. Genome wide $\pi_s$ refers to average pairwise nucleotide diversity at synonymous sites. SE: Southeast. **b** Principal coordinate analysis (PCA) based on genome-wide nucleotide diversity data. Average pairwise nucleotide diversity ($\pi$) calculated from all sites is shown for each population. **c** Decay of linkage disequilibrium (LD) with physical distance in four populations. The dashed line indicates an LD value of $r^2 = 0.33$. Data are deposited in figshare[58]

### Table 1 Summary of the sequencing data and detected mutations

| Sample ID | Treatment | # Mutations | Callable sites (Mb) |
|---|---|---|---|
| A | Indoor | 0 | 126.4 |
| E | Indoor | 0 | 125.7 |
| I | Indoor | 0 | 126.0 |
| J | Indoor | 0 | 126.4 |
| N | Indoor | 0 | 125.9 |
| B | Outdoor-noUV | 0 | 126.1 |
| G | Outdoor-noUV | 1 | 126.3 |
| K | Outdoor-noUV | 0 | 124.2 |
| O | Outdoor-noUV | 0 | 125.9 |
| P | Outdoor-noUV | 0 | 126.4 |
| C | Outdoor-UV | 1 | 125.6 |
| D | Outdoor-UV | 1 | 126.3 |
| L | Outdoor-UV | 1 | 126.3 |
| M | Outdoor-UV | 0 | 126.3 |
| Q | Outdoor-UV | 0 | 126.0 |

Each row shows the sample information and number of verified mutations. Effective sites are estimated as the total number of sites with sufficient coverage for finding de novo variants using our pipeline. The mutation rate is calculated as $\mu = $ (number of mutations/sum of effective sites)/number of generations. The average mutation rates (95% confidence interval) for samples grown under indoor, outdoor-noUV and outdoor-UV conditions are: $<7.92 \times 10^{-11}$ (NA), $7.92 \times 10^{-11}$ ($2.07 \times 10^{-11}$ to $3.98 \times 10^{-10}$), and $2.38 \times 10^{-10}$ ($4.76 \times 10^{-11}$ to $7.30 \times 10^{-10}$), respectively. The 95% confidence intervals were calculated based on the assumption that the number of mutations is Poisson distributed

Further analysis that compared the heterozygous sites of maternal and offspring individuals suggested a low false-negative rate for our mutation identification pipeline (average: 1.6%, 95% CI 0.2–3.0%). In addition, we spiked 1000 synthetic non-reference mutations to the sequence alignments and successfully recalled 945 of them using the same variant calling and filtering method. This gave a false negative rate of 5.5% (95% CI 4.1–7.2%). It is possible that we failed to detect mutations in regions of the genome not accessible using the current sequencing technology (mainly repetitive sequences). Given that the protein-coding region of the *S. polyrhiza* genome is 17.4 Mb, we estimate the number of mutations per generation in the entire protein-coding DNA of *S. polyrhiza* under natural, outdoor conditions to be $0.0041 \pm 0.0038$ (mean ± SD). As so few mutations were observed, we were unable to perform robust statistical analysis to test for treatment effects. However, the higher number of mutations found in outdoor samples and in the presence of UV light is consistent with the hypothesis that outdoor environmental factors increase the spontaneous mutation rate.

The genome-wide mutation rate in *S. polyrhiza* is within the range of mutation rates reported for unicellular eukaryotes and Eubacteria, but is more than seven times lower than the reported rates for multicellular eukaryotes (Fig. 2). This estimated seven-fold difference between *S. polyrhiza* and other multicellular eukaryotes is a conservative estimate, as all MA experiments in other organisms were performed under controlled indoor conditions, under which no mutations were observed in *S. polyrhiza*.

Based on these independent estimates of genetic diversity and mutation rate, we can estimate $N_e$ in *S. polyrhiza*. Assuming that mutation rates during the clonal and sexual reproduction phases of *S. polyrhiza* are equal, the estimated effective population size of *S. polyrhiza* is $9.8 \times 10^5$, which is among the highest estimates for multicellular eukaryotes, where $N_e$ was estimated using a similar approach (Supplementary Data 3). This method to estimate $N_e$, which is widely used, assumes populations are at their genetic equilibrium. Although there was little evidence from this study to suggest that the populations deviated dramatically from equilibrium conditions, as genome-wide Tajima's *D* estimates for three out of the four populations (Indian, European, and SE Asian) were close to 0 (in the American population Tajima's *D* was 0.59. Supplementary Table 3), more data is required to accurately evaluate the demographic history of *S. polyrhiza*.

The relatively large $N_e$ may have contributed to the evolution of a low mutation rate in *S. polyrhiza*, as selection can effectively drive down the mutation rate in populations with large $N_e$[8]. In

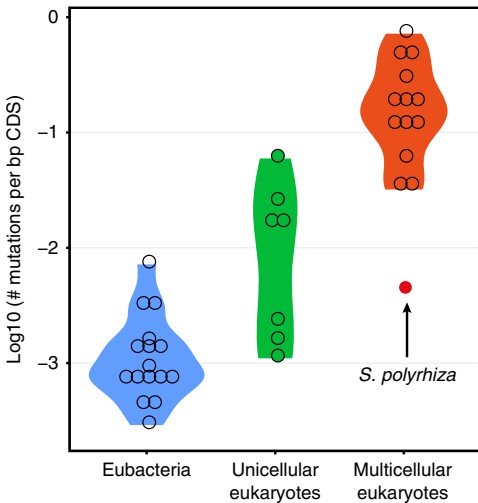

**Fig. 2** Estimated mutation rates in protein-coding regions among different organisms. The violin plots of $\log_{10}$-transformed numbers of mutations per base pair of protein-coding genome sequences (CDS) per generation for eubacteria, unicellular eukaryotes and multicellular eukaryotes, respectively. The kernel probability density is shown. Each circle indicates the estimate for one species. The arrow highlights the mutation rate in *S. polyrhiza*. Except for the mutation rate in *S. polyrhiza*, the plotted data were extracted from previous studies (Supplementary Data 3)

addition to large $N_e$, the infrequent sexual reproduction in *S. polyrhiza* might have enhanced the efficiency of selection to minimize the mutation rate. The relatively slow decay of genome-wide LD and the observation that *S. polyrhiza* rarely flowers in nature suggest that sexual reproduction and recombination in *S. polyrhiza*, at least in some populations, is infrequent relative to clonal reproduction. Such relatively infrequent recombination can increase linkage between a mutator allele and deleterious mutations, which as a consequence will enhance the strength of selection against mutator alleles[22]. Therefore, for species such as *S. polyrhiza* that reproduce both clonally and sexually, the frequency of asexual reproduction may be negatively correlated with the mutation rate. As ~80% of all angiosperms[23], including many crop species[24], can reproduce clonally, variation in the frequency of sex may have large effects on the evolution of mutation rates in plants and contribute to variation in intraspecific genetic diversity among species.

In addition to low mutation rate, linked selection can also reduce neutral nucleotide diversity by strong selection against deleterious alleles (background selection) or by substitutions of beneficial alleles at linked loci (genetic hitchhiking)[25], especially in species with relatively large LD[26,27]. However, to precisely estimate the effect of linked selection on reducing neutral diversity in *S. polyrhiza*, it is essential to characterize the factors that affect linked selection, such as the frequency of sweeps, the recombination rate, the strength of selection, the age of the beneficial allele, and population demography[28–31]. Future studies that sequence larger natural populations of *S. polyrhiza*, perform long-term outdoor selection experiments and simulate different demographic models will further shed light on the extent to which linked selection might have additionally contributed to the observed low genetic diversity in *S. polyrhiza*.

In this study, we show that the unexpected pattern of low genetic diversity in a species with large $N_e$ can be explained by extremely low mutation rate in *S. polyrhiza*. Linked selection may have further reduced the genome-wide genetic diversity. The role of mutation rate in driving variation in genetic diversity has been largely ignored, because obtaining accurate estimates of genome-

wide genetic diversity and spontaneous mutation rate in a range of organisms has been difficult in the past. Our study emphasizes that accurate estimates of mutation rates are important for explaining patterns of genetic diversity within species.

## Methods

**Mutation accumulation (MA) experiments with S. polyrhiza.** We performed a MA experiment with *S. polyrhiza* for 20 generations. *Spirodela polyrhiza* plants were propagated under three conditions: (i) indoors in the absence of UV light, (ii) outdoors in the absence of natural UV light, and (iii) outdoors in the presence of natural UV light. *Spirodela polyrhiza* genotype 7498 was pre-cultivated for three weeks in N-medium—which supports optimal growth (N-medium: 0.15 mM $KH_2PO_4$, 1 mM Ca $(NO_3)_2 \times 4\ H_2O$, 8 mM $KNO_3$, 5 μM $H_3BO_3$, 13 μM $MnCl_2 \times 4\ H_2O$, 0.4 μM $Na_2MoO_4 \times 2\ H_2O$, 1 mM $MgSO_4 \times 7\ H_2O$, 25 μM FeNaEDTA)—in a climate chamber operating under the following conditions: 16 h light, 8 h dark; light supplied by vertically arranged neon tubes (OSRAM, Lumilux, cool white L36W/840) on each side; light intensity at plant height: $186 \pm 3\ \mu mol\ s^{-1}\ m^{-2}$ outside polystyrene tubes and $142 \pm 3\ \mu mol\ s^{-1}\ m^{-2}$ inside polystyrene tube; temperature: 28 °C constant; humidity: 41%. The genotype 7498 originating from North Carolina (USA) was selected based on the existence of a clone-specific reference genome[14]. A single frond (S1) was transferred to a transparent 50 ml polystyrene tube ($28.5 \times 95$ mm, Kisker) containing 30 ml N-medium, covered with foam cap and incubated in a climate chamber under the above specified conditions. To obtain 6 MA lineages per treatment, the S1 ancestor was propagated according to the propagation scheme (see Supplementary Fig. 2) every two to three days when daughter fronds had fully emerged from the mother frond. For the indoor MA lines, 6 lineages were consequently propagated as single descendants for 20 generations under the same conditions as described above over a period of six weeks. For the outdoor MA lines, plants were moved at the end of June 2016 into a sun-exposed field site in Jena, Germany (50°53′06.7″N 11°40′53.1″E). The fronds were propagated in plastic beakers containing 180 ml N-medium that were fitted into the cavities of white polyvinyl chloride inserts (3 mm thickness) floating inside water-filled 10 l buckets. The buckets were surrounded with a 20 cm isolation layer of soil to avoid extreme temperature fluctuations and refilled with water to compensate for evaporating water whenever needed. To manipulate UV light, the buckets were covered with either UV transmitting (GS 2458, Sandrock, Germany) or UV blocking (UV Gallery100, Sandrock, Germany) Plexiglas plates with 1–3 cm distance between the bucket edge and the plates to allow air circulation. Each MA lineage was propagated in a separate bucket. After transplanting the fronds into the field, the buckets were shaded with two layers of green clear film for the first two days to allow plants to acclimate to outdoor conditions. The first green clear film layer was removed after two days, the second layer after four days. Plants were then propagated every 2–4 days for the following 2 months as single descendants for 20 generations. The MA lineages were randomized between the buckets every two weeks. The 20th generation of the outdoor plants was moved back to the original growth chamber. To obtain genomic DNA for whole genome re-sequencing (WGS), a single frond of the 20th generation of each of the indoor and outdoor MA lines and the ancestor, of which the roots and reproductive pockets were removed, was frozen in liquid nitrogen. All samples were stored at −80 °C until DNA extraction.

**DNA isolation and whole genome resequencing.** The plant tissue was ground by vigorously shaking the Eppendorf tubes with three metal beads for 1 min in a paint shaker (Skandex S-7, Fluid Management, Sassenheim Holland) at 50 Hz. All DNA samples were isolated using the CTAB method[32] and their quantity and quality was analyzed on Qubit. The DNA samples from the MA experiments were sequenced on Illumina HiSeq 4000 at the Genomics Center of the Max Planck Institute for Plant Breeding Research in Cologne (Germany) with 150 bp paired-end reads. For the 68 *S. polyrhiza* genotypes, all genotypes of *S. polyrhiza* (see Supplementary Data 1) were taken from the stock collection of the Matthias Schleiden Institute – Plant Physiology, University of Jena, Germany. Plants were then grown in N-medium (see details above) under a constant temperature of 28 °C and 41% humidity. Detailed information and origin of the 68 *S. polyrhiza* genotypes is listed in Supplementary Data 1. The genomes of the 68 genotypes of *S. polyrhiza* were sequenced on Illumina HiSeq X Ten at BGI (Shenzhen, China) with 150 bp paired-end reads. On average, 48.2 million reads per genotype were generated.

**Short-read trimming, mapping, and variant calling.** For all sequenced short reads, low-quality reads and adapter sequences were trimmed with Adapter-Removal v2.0[33] with the parameters: –collapse –trimns –trimqualities –minlength 36. All of the trimmed reads were then mapped to the *S. polyrhiza* reference genome[14] using BWA-MEM[34] with default parameters. All reads with multiple mapping positions in the genome were removed and only the mapped reads were kept. PCR duplicates were removed using the "rmdup" function from SAMtools[35]. The aligned reads were then used for variant (SNPs and small indels) calling using GATK v3.5[15] following the suggestions on best practices[36,37]. In brief, the aligned reads around indels were re-aligned using "IndelRealigner", and variants were called using the UnifiedGenotyper function with the option -stand_call_conf 30 -stand_emit_conf 10. The variants were then filtered with the option MQ0 ≥ 4 && ((MQ0/(1.0 × DP)) > 0.1) & QUAL < 30.0 & QD < 5.0, which removes all variants that either have more than four samples with MappingQualityZero (MQ0, low

mapping quality) and 10% of the mapped reads (DP) with low mapping quality, or have low Phred-scaled probability that a polymorphism exists at the site (QUAL), or low qual score normalized by allele depth (QD). The variant clusters were further annotated as more than three variants within 50 bp using the GATK VariantFiltration function. Only biallelic loci were kept for downstream analysis. The synonymous and non-synonymous variants were annotated using snpEFF (version 4.3 m)[38]. Due to low sequencing coverage, three individuals from the MA experiments were removed from downstream analysis (see Supplementary Fig. 2).

**Population genomic analysis.** To analyze genetic diversity and population genomics of the 68 genotypes, additional filtering steps were performed using vcffilter ([https://github.com/vcflib/vcflib], with parameters: -s -f DP > 510 & DP < 10,200). Variants from mitochondrial and chloroplast regions and clustered variants were removed using vcftools[39]. The population structure among the sequenced 68 genotypes was analyzed using fastSTRUCTURE v1.0[40]. To this end, the loci that were not in Hardy-Weinberg equilibrium ($P < 0.01$) and tightly linked loci ($r^2 > 0.33$) were removed using vcftools and bcftools[41], respectively. Multiple $K$ values (refers to number of populations) ranging from 1 to 10 were analyzed and the value $K = 4$ was selected using the chooseK.py function from the fastSTRUCTURE package. The genome-wide intraspecific diversity was analyzed using Popgenome v2.2.0[42] using a data set with no missing genotypes (61,281 SNPs, ~5% of total SNPs, were removed), and diversity at synonymous and non-synonymous sites was analyzed using SNPGenie[43] using all variants. Overall, more than 88.0% of the genome and ~92.8% of the coding region had sufficient coverage for variant calling. The estimated population genomic summary statistics were then corrected based on the callable sites. Plink[44] was used to calculate pairwise linkage disequilibrium (LD) from the dataset, for which related individuals were removed and only SNPs with MAF greater than 0.05 were kept. To model the decline of LD with physical distance, pairwise $r^2$ between sites was used as the use of $D'$ is sensitive to small sample sizes[45,46], and the decline of LD was modeled using Sved's equation: $E(r^2) = (1 - /(1 + 4\beta d)) + 1/n$, where $\beta$ is the decline in LD with distance $d$[47] and $1/n$ accounts for small sample size[48]. The extent of useful LD for mapping can be defined as $r^2 = 0.33$[49]. In this study we use mean $r^2$ for non-overlapping 100-bp bins to fit Sved's equation.

**Mutation rate estimation and false-negative calculations.** Accurately estimating mutation rate requires a step-wise filtering and quality checking process. The SNP filtering pipeline for the MA experiments was developed based on previous studies[50,51] and iterative manual inspections of the BAM files using Integrative Genomics Viewer (IGV)[52,53]. (1) To reduce false positives, we only considered the mapped and properly paired reads with insertion size greater than 100 bp and less than 600 bp using bamtools[54]. (2) We also excluded all genomic regions that were supported by fewer than nine or greater than 75 reads per sample from both variant counting and genome size calculation, as the variants from the regions that have low or high coverage are likely due to mapping errors (such as repetitive or duplicated regions). On average, 79.7% of the genomic region was kept. (3) Because spontaneous mutations should be only found in the offspring samples but not the ancestor, and the likelihood of a mutation occurring at the exact same position in multiple samples is extremely low ($u^n$, where $u$ is the mutation rate, and $n$ refers to number of samples that have a mutation at the same position), any variants that appeared in more than two samples were removed. (4) Only the heterozygous variants that were supported by at least three reads for both alleles were kept. After these filtering steps, 86 variants were found (Supplementary Data 5). Among these, 56 were annotated as variant clusters, likely due to mapping errors. To confirm this, we re-sequenced 28 of these variants that were located in clusters using a Sanger sequencing approach and found none of them confirmed to be true mutations. Therefore, all the variants that were classified as variant clusters were removed.

After removing all variant clusters, nine SNPs and 21 indels remained. Among the 21 indels, all of them were loss of heterozygosity in either the ancestral or the offspring samples. Inspecting the alignment using the IGV showed that 19 of them were located in regions of simple sequence repeats or transposable elements, which were likely false positives. To confirm this, we selected 11 indels for Sanger sequencing and found that all of them were indeed false positives. As a result, all 21 indels were removed from the downstream analysis. Among the nine SNPs, six were point mutations (due to spontaneous mutations) and three were loss-of-heterozygosity (LOH) mutations (potentially due to gene conversion events). We further validated these SNPs using a Sanger-sequencing approach. Two LOH loci were very close to the gap of the genome assembly and the PCR primers could thus not be designed. We validated the remaining seven loci (six point-mutations and one LOH). In total, four out of the six point-mutations were confirmed, and the loss of heterozygosity mutation turned out to be a false positive. The confirmed point-mutations are listed in Table 1 and were used for calculating the spontaneous mutation rate.

The relatively stringent parameters in the variant filtering process theoretically could result in a high rate of false negatives. To control this, we further estimated the false negative rate using the sequence data. We first identified all high-quality heterozygous SNP loci (30,392) from the ancestor using the same filtering parameters (coverage between 9 and 75, and at least three reads to support each of the reference and the alternative allele) and compared them with the heterozygous SNPs in the offspring using a custom script. In theory, all these variants should be found in the clonally produced offspring. Thus, the number of SNPs that could not be identified from the offspring was used to estimate the highest boundary of the

false negative rate from our sequencing and variant calling/filtering pipeline, as some of these cases could be a true loss of heterozygosity.

In addition, we also estimated the false negative rate by simulating synthetic mutations to the sequence alignments, an approach that has been used previously[55,56]. We introduced 1000 non-reference mutations to the callable regions using BAMsurgeon[57] (with parameter: –mindepth 9 –maxdepth 75 -d 0 –aligner mem –insane –force), with a frequency of 0.5 (standard deviation = 0.1). Using the same variant calling and filtering pipeline, we identified 94.5% (945 out of 1000) of the synthetic mutations that were successfully introduced to the BAM files, yielding an average false negative rate of 5.5% (95% CI 4.1–7.2%).

**Variant validation using Sanger sequencing.** Because the total amount of DNA from a single individual was limited, the variant validation was performed using the descendants of the ancestor and offspring individuals. Specifically, at the end of the MA experiments, one individual of each line was propagated for four more generations under indoor conditions, after which the plants were frozen in liquid nitrogen for subsequent variant validation.

To validate the candidate variants, DNA was isolated as described above. PCR primers were designed based on the 500 bp flanking sequences. The PCR reactions were performed with goTaq DNA polymerase (Promega) using 30 PCR cycles with an annealing temperature of 58 °C. The primer information is listed in Supplemental Data 6. The PCR products were checked on a 1.5% agarose gel. The PCR products were then used for sequencing reactions using BigDye v3.1, and the products from the sequencing reactions were purified and sequenced on an ABI 3130XL sequencer.

**Reporting summary.** Further information on experimental design is available in the Nature Research Reporting Summary linked to this article.

## Data availability
All raw DNA sequences obtained in this study are submitted to NCBI under Bioproject PRJNA476302. Data for figures are deposited in figshare at https://doi.org/10.6084/m9.figshare.7599767.v1 (ref. [58]). The authors declare that the data supporting the findings of this study are available within the article, its Supplementary Information files, and upon request.

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

## Acknowledgements
We thank Claudia Michel, Beatrice Arnold and Yuanyuan Song for their help in validating the variants from the MA experiments, Stefanie Schirmer for help with the MA experiment and DNA isolation, Daniel Veit for manufacturing the facilities for outdoor duckweed growth, Thomas Städler and Martin Schäfer for constructive discussions commenting on the manuscript. We are also grateful to Tobias Neumann for providing meteorological data. This work was supported by a Marie Curie Intra-European Fellowship (No: 328935 to S. X.), the Alfred and Anneliese Sutter-Stöttner Foundation (to S. X. and M. H.), the Center for Adaptation to a Changing Environment (ACE) at ETH Zurich (to S. X., J. S. and A. W.), the Max Planck Society and the University of Münster. We acknowledge support from the Open Access Publication Fund of the University of Muenster.

## Author contributions
S. G., A. W. and M. H. performed the experiments, J. S., J. B. and S. X. performed data analysis, K. J. A. and K. S. S. contributed to the giant duckweed collections, A. W. and J. G. provided resources. M. H. and S. X. conceived and supervised the project, S. X. wrote the manuscript with input from all co-authors.

## Additional information

**Competing interests:** The authors declare no competing interests.

