## [Peer Review File · Nature Communications]

Reviewers' Comments:

Reviewer #1:

Remarks to the Author:

The authors report a study in which they (1) directly estimate the spontaneous mutation rate by WGS of a set of 15 mutation accumulation (MA) lines, (2) estimate the standing genetic variation (π) from average pairwise heterozygosity at silent sites in a worldwide sample of 68 clones, and (3) use the resulting estimate of the base-substitution rate u to infer the effective population size from the relationship $\theta(\pi) = 4N_e u$. The key result is that the base substitution rate is very low for a multicellular eukaryote, as is the average pairwise heterozygosity, leading to an inferred very large N_e . From these results, they conclude that the large N_e ($\sim 10^6$) has resulted in efficient selection to reduce the mutation rate to a low level, consistent with the "drift barrier" hypothesis of mutation rate evolution.

Overall, this is a very nice study, with enviously clear results. In some sense it constitutes just another brick in the wall of science, but it seems like a pretty solid brick.

I have a few concerns/comments:

1. First, it's not completely clear what criteria were used to include or exclude sites from the analysis. On line 237 they say: "...only SNPs with minor allele frequency (MAF) greater than 0.05 were kept...". Does that apply only to the LD calculation, or to the calculation of average pairwise heterozygosity. Comparisons to published data will be meaningful only if all of the estimates were calculated in the same way.
2. The method for detecting false negatives is pretty convincing, and I believe the 1.6% error rate, as far as it goes. However, the $\sim 20\%$ of the genome that is not included in the analysis is probably the most mutation-prone. At least some of the taxa to which duckweed is compared have a much larger fraction of the genome included in the mutation analysis. I don't see an easy way around this problem, but it should be acknowledged. I do see a hard way around this problem, which is to calculate the mutation rate for each taxon at only silent sites, or all coding sites. But I sure wouldn't want to do that.
3. Given the very small number of mutations expected per genome (about 2.5, I think), I want to see the upper confidence interval on the inferred mutation rate. Clearly the mutation rate is low, but how high could it be and still be consistent with the data? Keightley et al. (2014, Genetics) reported confidence intervals on the base sub rate inferred from six mutations in 12 individuals.
4. The fact that the point-estimate of the base-sub rate is seven-fold less than the average for multicellular Euks is interesting and important. However, recall that Ness et al. (2015, Genome Res) documented a seven-fold difference in mutation rate among six genotypes of *Chlamydomonas*, one of which was well into the range of multicellular Euks.
5. Finally, it would be interesting to see how Tajima's D varies among the four sub-populations. That would help put the different π 's in context.

Reviewer #2:
Remarks to the Author:
Review of Xu et al.,

This is a seemingly straight-forward population-genomic and mutational analysis of a small-sized and globally abundant plant species, the aquatic duckweed. The authors combined a global survey of genome-wide nucleotide diversity with results from a mutation-accumulation experiment to show that this organism has an exceptionally low mutation rate per nucleotide site per generation, combined with a very large effective population size. The combination of these two factors is manifest in the very low silent-site diversity found in the species – remarkably low, particularly given that this applies even with a global survey.

Overall, the results are quite consistent with the drift-barrier hypothesis (DBH), which predicts an evolutionary decline in the mutation rate as N_e increases and selection becomes more efficient. This is an especially nice study because it extends this theory well beyond the domain for which there are data for multicellular species.

Specific comments:

- 1) The authors state that low nucleotide diversity in populations with large N_e is a “counter-intuitive” prediction of the drift-barrier hypothesis. However, this could be viewed as a bit of an overstatement. In the extreme, for example, the DBH might predict a perfectly inverse relationship between N_e and the mutation rate, in which case the standing level of variation would be completely independent of the mutation rate. I don’t think things are quite that extreme, but I do think it is well appreciated that variation will not scale linearly with N_e under the DBH.
- 2) The statement in the paper on LD (lines 94--) is not very clear, as no information is given on the change in LD with distance between sites. Such information is actually in the figure, and this should be referred to more directly in the text. It should also be noted that there is a mathematical bound on the values that r^2 can take, which is a function of allele frequencies, so it is unclear whether the differences among geographic regions is simply a function of differences in the site-frequency spectrum. In addition, the traditional interpretation of the relation of r^2 with physical distance (mentioned in the methods section) doesn’t entirely hold up to scrutiny (see Walsh and Lynch book), and in any event applies to the pattern within a single population (and not in any clear way (at least to me) to a mixture of samples from a subdivided population).
- 3) As the authors have data on their ancestral mutation-accumulation lines, and the latter were propagated in an asexual manner, it ought to be possible to look for loss of heterozygosity owing to gene conversion (revealed as heterozygous sites becoming homozygotes). Not absolutely essential for this paper, but would certainly be nice information to have. (Although such activity would change within clone variation, it should not have implications for the estimates of total diversity in nature).
- 4) The discussion on recombination and the evolution of the mutation rate is a bit garbled. The basic idea that the strength of selection on the mutation rate is diminished by recombination goes back to the classical work of Kimura and was taken up in more detail by Lynch in a Genetics paper.
- 5) The grey dots in Figure 2 are essentially invisible, and the meaning and basis of the kite diagrams is unclear.

Reviewer #3:

Remarks to the Author:

In the paper. "Title" Author et al combine population genomic diversity data with a mutation accumulation experiment to explore mutation rate and effective population size of duckweed. The paper addresses a fundamentally interesting question and goes beyond the norm in combining these two datasets to make new insights into mutation rate evolution. Although I find the paper very interesting I think there are a conceptual and methodological problems that I think could really change the paper. Specifically if the mutation rate estimate is biased and too low, diversity is not in equilibrium and Lynch's theory does not predict declining diversity with increased N_e , the results will be very different. I would need to see the below issues addressed before I could trust the results and recommend publication. I outline those concerns and highlight a few minor issues that might improve the paper

Major Comments

1. Can μ decrease with higher N_e 's to the point where π is reduced?

The authors state: "However, one counter-intuitive prediction of the drift-barrier hypothesis is that populations with large N_e may also have low genetic diversity if natural selection has driven mutation rates to very low levels" - Lynch's hypothesis doesn't predict large N_e populations would have low diversity - just lower than diversity would be if it linearly increased with N_e . i.e. Diversity may plateau. This is both a wording problem and a fundamental problem with the explanation provided - large N_e drives down μ , but can μ be driven down to the point where diversity actually declines? There are other species where N_e is much higher and μ is similar or even higher and diversity is much higher - I think Lynch's original model should be able to show if this is a realistic prediction. I think checking whether the results are consistent with the theory needs to be more than a verbal model and might require some math.

2. Estimate of the mutation rate denominator

The key finding to this paper is that the mutation rate is extremely low. Of course proving low mutation rates is hard because you're effectively proving a negative. I feel confident that there are few false positive mutations - but the method for assessing false negatives is not good and is likely to be highly biased - in effect the authors measures the rate of missed calls at the highest quality sites in the parental genomes. The correlation of how well mapped a region would be between the parent and its offspring will be incredibly high. Keightley et al (2015) devised a more robust false negative screen - where mutations are simulated and the rate at which these known mutations can be identified is recorded. False negatives are sites where the mutation was not called but end up contributing to the "total" number of sites in the denominator of mutation rate calculations. The difficulty here is not having a known set of heterozygote sites to get an unbiased estimate of the allele depths at heterozygous sites.

3. Can you discuss whether diversity ($4N_e\mu$) is at equilibrium.

Diversity only reflects N_e and μ when at equilibrium - if the population has recently changed size, or selective sweeps are common in asexual populations the diversity seen may not reflect these patterns. There are a few common ways to look at this that might help:

- What is " π_W " - Is this Watterson's theta - If there is an excess of rare variants it would show up in a discrepancy of Watterson's theta and π , or in the site frequency spectrum - ie Tajima's D
- Can the authors fit a simple population expansion model to estimate N_e ?
- What is the denominator for π calculations (ie per what site)
- Why is π_{intron} and $\pi_{intergenic}$ not reported? Is it comparable to π_s ?

Minor Comments

L38 The sentence is basically repeating itself

L47 You say its 'controversial' twice in this paragraph

L56 "Whether this pattern is present in eukaryotes is unknown, largely due to the paucity of studies quantifying both genome-wide diversity and spontaneous mutation rates under natural conditions in organisms with different life histories and reproductive strategies." Its unclear from what is written why natural conditions nor life history are relevant here - can the authors provide citations and logic to back this?

L86 "mis-labeling during long-term maintenance of the duckweed collections" - is this a problem - Does it change any of your results - especially diversity?

L93 - "distance between variants when their correlation coefficient (r^2) = 0.2" where does this cutoff for estimating LD plateau come from? It seems arbitrary and very high

L103 - "Mutation rates can be markedly affected by outdoor environmental stresses such as temperature fluctuations and ultraviolet (UV) light 16-20" — These are strange references - as the UV experiments are artificial (not outdoor conditions) and the study on Epistasis is theory about selection and not relating to spontaneous mutation rates. - the premise that duckweed is stressed by environment and UV is a bit of a straw man argument. An organisms outdoor environment is what its been adapting too and there is no evidence provided it is "stressed" - I don't think you can say that mutation rate is likely lower in MA experiments. For many organisms not optimized for growth in the lab the conditions may be more stressful, without evidence - none of which is provided in this paper - can we say this is true.

L481 - "referes" - typo

It is unclear from the experimental description how the selection of fronds occurred between generations - A single frond was taken to start a new colony - but how was that frond selected - there is a chance for selection to enter the experiment if fronds with low growth rate are ignored, or those that accumulate mutations are less likely to contribute to the next generation. If the production of fronds is not synchronized how confident are the authors about the number of generations per transfer

L221 - "220. ""MQ0 >= 4 && ((MQ0 / (1.0 * DP)) > 0.1) & QUAL < 30.0 & QD < 5.0"" - Can this be unpacked into normal words.

L229 Expand on - "Loci with missing data" how many missing sites were required before the site was removed - how many sites did this affect? How many invariant sites are affected?

L233 - For structure were loci sub-sampled? Linked sites should not be used to infer structure

Throughout "Intra-specific" - is one word no hyphen, intraspecific

"Related individuals were removed and only SNPs with minor allele frequency (MAF) greater than 0.05 were kept" - does this apply only to the plink LD analysis

L241 - "The extent of useful LD for mapping can be defined as $r^2 = 0.240$ " - where does this statement come from?

L136 - "This relatively large N_e may have contributed to the evolution of a low mutation rate in *S. polyrhiza*, as selection can effectively drive down the mutation rate in populations with large N_e " The N_e is indeed large for a eukaryote but not for a unicellular organism - many of which have higher mutation rates despite having genomes dense with functional sequence and also high diversity. This relates to my "major" point above.

143-148 I think the author's thinking is backwards - if recessive mutations are sheltered, how can they drive the mutation rate to evolve lower? I would expect less selection on a mutator allele.

L255 " U^n where U is the mutation rate, and n refers to number of samples" I think this is not the correct equation. There are multiple generations and you only need two samples to have the mutation, not all n

Reviewers' comments:

Reviewer #1 (Remarks to the Author):

The authors report a study in which they (1) directly estimate the spontaneous mutation rate by WGS of a set of 15 mutation accumulation (MA) lines, (2) estimate the standing genetic variation (π) from average pairwise heterozygosity at silent sites in a worldwide sample of 68 clones, and (3) use the resulting estimate of the base-substitution rate u to infer the effective population size from the relationship $\theta(\pi) = 4N_e u$. The key result is that the base substitution rate is very low for a multicellular eukaryote, as is the average pairwise heterozygosity, leading to an inferred very large N_e . From these results, they conclude that the large N_e ($\sim 10^6$) has resulted in efficient selection to reduce the mutation rate to a low level, consistent with the "drift barrier" hypothesis of mutation rate evolution.

Overall, this is a very nice study, with enviously clear results. In some sense it constitutes just another brick in the wall of science, but it seems like a pretty solid brick.

Thanks for this nice summary.

I have a few concerns/comments:

1. First, it's not completely clear what criteria were used to include or exclude sites from the analysis. On line 237 they say: "...only SNPs with minor allele frequency (MAF) greater than 0.05 were kept...". Does that apply only to the LD calculation, or to the calculation of average pairwise heterozygosity. Comparisons to published data will be meaningful only if all of the estimates were calculated in the same way.

Thanks for raising this point. We did not apply a MAF filter to data used for calculating the average pairwise heterozygosity. A MAF filter (>0.05) was applied to the data for the estimation of linkage disequilibrium because low frequency alleles can upwardly bias LD¹. There is no consensus on a MAF cutoff for estimating of LD, but 0.05 is commonly used^{2,3}. In the revision, we clarified this point in the method section. See lines 257-258.

2. The method for detecting false negatives is pretty convincing, and I believe the 1.6% error rate, as far as it goes. However, the ~20% of the genome that is not included in the analysis is probably the most mutation-prone. At least some of the taxa to which duckweed is compared have a much larger fraction of the genome included in the mutation analysis. I don't see an easy way around this problem, but it should be acknowledged. I do see a hard way around this problem, which is to calculate the mutation rate for each taxon at only silent sites, or all coding sites. But I sure wouldn't want to do that.

Indeed, to measure the mutation rate in the repetitive region is extremely challenging. We added a sentence in the results and discussion section to fully acknowledge this challenge. See lines 124-126.

3. Given the very small number of mutations expected per genome (about 2.5, I think), I want to see the upper confidence interval on the inferred mutation rate. Clearly the mutation rate is low, but how high could it be and still be consistent with the data? Keightley et al. (2014, Genetics) reported confidence intervals on the base sub rate inferred from six mutations in 12 individuals.

We adopted the method from Keightley et al (2014, and 2015) to estimate false negatives and confidence intervals of the mutation rate in *S. polyrhiza*. In brief, we randomly selected 1000 positions (with coverage between 9 and 75) in the genome and introduced synthetic non-reference mutations to the bam files using BAMSurgeon⁴. Among them, 51 positions showed low coverage and were therefore dropped. Among the 949 successfully introduced mutations, 931 were identified using our pipeline (97.2%), which gave a similar false positive rate to the one we previously estimated. The missed variants were identified at the variant calling stage, but were removed at the filtering stage as they were found in SNP clusters.

Furthermore, using the method from Keightley et al 2015, the 95% confidence interval of the mutation rate in *S. polyrhiza* grown outdoors is: $4.76 \times 10^{-11} - 7.30 \times 10^{-10}$, assuming the number of mutations is Poisson distributed. We included the additional false negative rate estimations in the main text and added the CI estimates in Table 1.

4. The fact that the point-estimate of the base-sub rate is seven-fold less than the average for multicellular Euks is interesting and important. However, recall that Ness et al. (2015, Genome Res) documented a seven-fold difference in mutation rate among six genotypes of *Chlamydomonas*, one of which was well into the range of multicellular Euks.

Thanks for pointing this out. The intra-specific variations of mutation rate in the haploid, unicellular *Chlamydomonas* is indeed interesting. However, even the lowest mutation rate among all examined *C. reinhardtii* genotypes is 4.05×10^{-10} , which is still slightly higher than the average mutation rate measured in *S. polyrhiza*.

Reviewer #2 (Remarks to the Author):

Review of Xu et al.,

This is a seemingly straight-forward population-genomic and mutational analysis of a small-sized and globally abundant plant species, the aquatic duckweed. The authors combined a global survey of genome-wide nucleotide diversity with results from a mutation-accumulation experiment to show that this organism has an exceptionally low mutation rate per nucleotide site per generation, combined with a very large effective population size. The combination of these two factors is manifest in the very low silent-site diversity found in the species – remarkably low, particularly given that this applies even with a global survey.

Overall, the results are quite consistent with the drift-barrier hypothesis (DBH), which predicts an evolutionary decline in the mutation rate as N_e increases and selection becomes more efficient. This is an especially nice study because it extends this theory well beyond the domain for which there are data for multicellular species.

Thanks for this nice summary.

Specific comments:

1) The authors state that low nucleotide diversity in populations with large N_e is a “counter-intuitive” prediction of the drift-barrier hypothesis. However, this could be viewed as a bit of an overstatement. In the extreme, for example, the DBH might predict a perfectly inverse relationship between N_e and the mutation rate, in which case the standing level of variation would be completely independent of the mutation rate. I don’t think things are quite that extreme, but I do think it is well appreciated that variation will not scale linearly with N_e under the DBH.

Thanks for raising this point. We fully agree with the reviewer that the relationship

between N_e and genetic variation does not scale linearly. Since the same point was also raised by another reviewer, we included our detailed response below, under the responses to the 1st comment from review 3. We also revised the introduction accordingly. See lines 48-58.

2) The statement in the paper on LD (lines 94--96) is not very clear, as no information is given on the change in LD with distance between sites. Such information is actually in the figure, and this should be referred to more directly in the text. It should also be noted that there is a mathematical bound on the values that r^2 can take, which is a function of allele frequencies, so it is unclear whether the differences among geographic regions is simply a function of differences in the site-frequency spectrum. In addition, the traditional interpretation of the relation of r^2 with physical distance (mentioned in the methods section) doesn't entirely hold up to scrutiny (see Walsh and Lynch book), and in any event applies to the pattern within a single population (and not in any clear way (at least to me) to a mixture of samples from a subdivided population).

We agree that the decline in LD with physical distance should not be calculated for data from structured/subdivided populations. In the manuscript we calculated the decline in LD with physical distance for each population separately and the range between the populations was provided in main text. We have revised the text to make this clearer (lines 93-96).

“The extent of linkage disequilibrium (LD) declined with physical distance between linked loci and this rate of decline varied between populations (Figure 1C). The average distance between SNPs with a LD correlation coefficient (r^2) = 0.33, ranged from 8.6 kb in the SE Asian population to 86.8 kb in the European population.”

We excluded loci with $MAF < 0.05$ to reduce potential bias in LD that may be introduced by low frequency alleles (see our response to Reviewer #1 comment1). The LD estimates are for loci on the same chromosome, and thus we have not estimated LD between unlinked loci. For loci on the same chromosome the LD will decline with distance and the rate of decline is influenced by many factors – predominantly recombination rate between two loci, selection and population demography. We are not aware of any evidence that disputes this relationship. We have reviewed the latest version of Lynch and Walsh “Evolution and Selection on Quantitative Traits” and we could not find passages that suggest this relationship does not hold up to scrutiny.

3) As the authors have data on their ancestral mutation-accumulation lines, and the latter were propagated in an asexual manner, it ought to be possible to look for loss of heterozygosity owing to gene conversion (revealed as heterozygous sites becoming homozygotes). Not absolutely essential for this paper, but would certainly be nice information to have. (Although such activity would change within clone variation, it should not have implications for the estimates of total diversity in nature).

We agree with the reviewer, analyzing gene conversion rates could give added value to our manuscript. We attempted such an analysis, but we could not determine if loss-of-heterozygosity events were due to gene conversion or due to false negatives when identifying the variants. Because the false negative rate estimations using simulations revealed similar false negative rate as the loss-of-heterozygosity events, the only way to gain high confidence on the estimates of gene conversion rate is to validate a substantial proportion of these putative gene conversion events using Sanger sequencing. However, because we have a very limited amount of DNA from a single frond (3-5mm diameter) it was not possible to perform many additional sequencing reactions on the same individual.

4) The discussion on recombination and the evolution of the mutation rate is a bit garbled.

The basic idea that the strength of selection on the mutation rate is diminished by recombination goes back to the classical work of Kimura and was taken up in more detail by Lynch in a Genetics paper.

Thanks for pointing this out and giving us a chance to clarify this. We have revised the text to make this discussion clear. See lines 149-157.

5) The grey dots in Figure 2 are essentially invisible, and the meaning and basis of the kite diagrams is unclear.

We now revised the Figure 2 by increasing the contrasts of the grey dots. The kite diagrams in the figure are the violin plots from the ggplot package, which depict the kernel probability density of the data. In the revision, we added this information in the figure legend. See changes in Figure 2 and its legend.

Reviewer #3 (Remarks to the Author):

In the paper. "Title" Author et al combine population genomic diversity data with a mutation accumulation experiment to explore mutation rate and effective population size of duckweed. The paper addresses a fundamentally interesting question and goes beyond the norm in combining these two datasets to make new insights into mutation rate evolution. Although I find the paper very interesting I think there are a conceptual and methodological problems that I think could really change the paper. Specifically if the mutation rate estimate is biased and too low, diversity is not in equilibrium and Lynch's theory does not predict declining diversity with increased N_e , the results will be very different. I would need to see the below issues addressed before I could trust the results and recommend publication. I outline those concerns and highlight a few minor issues that might improve the paper

Major Comments

1. Can μ decrease with higher N_e 's to the point where π is reduced?

The authors state: "However, one counter-intuitive prediction of the drift-barrier hypothesis is that populations with large N_e may also have low genetic diversity if natural selection has driven mutation rates to very low levels" - Lynch's hypothesis doesn't predict large N_e populations would have low diversity - just lower than diversity would be if it linearly increased with N_e . i.e. Diversity may plateau. This is both a wording problem and a fundamental problem with the explanation provided - large N_e drives down μ , but can μ be driven down to the point where diversity actually declines? There are other species where N_e is much higher and μ is similar or even higher and diversity is much higher - I think Lynch's original model should be able to show if this is a realistic prediction. I think checking whether the results are consistent with the theory needs to be more than a verbal model and might require some math.

Thanks for this suggestion. The DBH predicts that effective population size (N_e) is negatively correlated with mutation rate (μ). Indeed, data from a diversity of different species showed significant correlations between $\log(\mu)$ and $\log(N_e)$ (Lynch et al. 2016):

$$\ln(\mu) = \alpha + \beta * \ln(N_e) \quad \text{Eq1.}$$

Here, α is the intercept, and β is the correlation coefficient between mutation rate and effective population size.

Since genetic diversity (θ) is jointly determined by N_e and mutation rate:

$$\theta = 4 * N_e * \mu \quad \text{Eq2.}$$

we can derive the equation

$$\theta = 4 * e^{(\alpha + \beta * \ln(N_e))} * N_e \quad \text{Eq 3.}$$

Transforming Eq3 gives:

$$\theta = 4 * e^{\alpha} * N_e^{(1+\beta)} \quad \text{Eq 4.}$$

Therefore, the relationship between N_e and genetic diversity (θ) depends on the value β , which is the correlation coefficient between $\ln(\mu)$ and $\ln(N_e)$. Because the correlation is negative, β is smaller than 0. In this equation, when $\beta > -1$, θ increases with N_e and when $\beta < -1$, θ decreases with N_e , and when $\beta = -1$, θ is independent of N_e (as mentioned by reviewer 1).

Using the data from both multicellular and single cellular organisms (from Lynch et al. 2016), we derived 99% CI of β : from -0.89 to -0.60. When we only consider the multicellular eukaryotes (including *S. polyrhiza*), we derived 99% CI of β : from -1.15 to -0.02. Therefore, in theory, the value of β can be smaller than -1, which will result in an unexpected pattern that a species with higher N_e has lower genetic diversity due to extremely low mutation rate.

In theory, the parameter β is affected by the strength of selection. For a given N_e , if the selection coefficient against increased mutation rate (Kimura 1966) is strong, selection will drive mutation rate to a very low level (β will be smaller). Several factors may affect the selection coefficient. For example, recombination is known to weaken the strength of selection on reducing mutation rate, and background selection can also increase the overall levels of selection. Therefore, in organisms that have low recombination rate and/or high background selection, purifying selection might drive mutation rate to very low level (e.g., resulting in $\beta < -1$).

We have revised the original text to avoid any confusion. See lines 48-58

2. Estimate of the mutation rate denominator

The key finding to this paper is that the mutation rate is extremely low. Of course proving low mutation rates is hard because you're effectively proving a negative. I feel confident that there are few false positive mutations - but the method for assessing false negatives is not good and is likely to be highly biased - in effect the authors measure the rate of missed calls at the highest quality sites in the parental genomes. The correlation of how well mapped a region would be between the parent and its offspring will be incredibly high. Keightley et al (2015) devised a more robust false negative screen - where mutations are simulated and the rate at which these known mutations can be identified is recorded. False negatives are sites where the mutation was not called but end up contributing to the "total" number of sites in the denominator of mutation rate calculations. The difficulty here is not having a known set of heterozygote sites to get an unbiased estimate of the allele depths at heterozygous sites.

Thanks for this suggestion. We now performed the suggested analysis. See details in the response to the 3rd point of reviewer 1. Overall, using the suggested method, we found a similar false negative rate as our previous estimate. The method and results are also included in the revised manuscript. See lines 122-126 and lines 308-314.

3. Can you discuss whether diversity ($4N_e\mu$) is at equilibrium.

Diversity only reflects N_e and μ when at equilibrium - if the population has recently changed size, or selective sweeps are common in asexual populations the diversity seen may not reflect these patterns.

The DBH hypothesis can explain variation in mutation rate across a diverse range of organisms with different mating systems and historical demographics (See Lynch et al. 2016). This suggests that this model is relatively robust to deviations from equilibrium. Estimates of Tajima's D do not indicate large deviations from neutrality, with the exception of the North American population, and inbreeding coefficients are within a normal range. However, we felt that without additional data that allow us to perform robust demographic modelling, it is

difficult to confidently demonstrate whether the population is at equilibrium. We added this information to the text in the revised version of the manuscript. See lines 142-148.

There are a few common ways to look at this that might help:

- What is " π_W " - Is this Watterson's theta - If there is an excess of rare variants it would show up in a discrepancy of Watterson's theta and π , or in the site frequency spectrum - ie Tajima's D

Thanks for this suggestion. We did estimate Tajima's D previously, and now we include this data in the revised MS, providing an estimate of Tajima's D for each population and for each chromosome. The Tajima's D values are overall close to 0, which indicate the populations are close to genetic equilibrium. See table S5.

- Can the authors fit a simple population expansion model to estimate N_e ?

Using coalescent simulations (the MS function from PopGenome), we can estimate Tajima's D and π under different demographic models and use this to estimate N_e . When we did this, the statistics were similar across these models - using π we can estimate N_e for a stable population $N_e=8.4*10^5$, expanding population $N_e=4.5*10^5$, declining population $N_e=3.3*10^5$. We could not statistically distinguish between different models using these summary statistics, which is probably due to the combined effects of small sample size and low genetic diversity⁵. Additional data would be needed to accurately estimate the historical demography of these populations, but this is outside the scope of the current study.

- What is the denominator for π_s calculations (ie per what site)

We calculated the π_s using SNPGenie⁶, in which the denominator for calculating the π_s is the total number of synonymous sites. Further analysis also showed that only 7.2% of the exon regions have too low or too high coverage (not callable). Therefore, using total number of synonymous sites is justified. Nevertheless, we now adjusted the estimation of π_s according to the number of callable sites. See changes in table S2.

- Why is π_{intron} and $\pi_{intergenic}$ not reported? Is it comparable to π_s ?

We already included the $\pi_{intergenic}$ in Table S2, which was named as $\pi_{noncoding}$. To avoid the confusion, we now named it as $\pi_{intergenic}$. In the revision, we also included π_{intron} in Table S2. Overall, $\pi_{intergenic}$ (0.0016) is higher than π_s and π_{intron} . See table S2.

Minor Comments

L38 The sentence is basically repeating itself

We revised this sentence. It now reads: "Explaining within-species genetic diversity—measured as the level of intraspecific DNA sequence variation—is one of the major goals in evolutionary and conservation biology^{1,2}".

L47 You say its 'controversial' twice in this paragraph

We now changed the word "controversial" to "largely unclear".

L56 "Whether this pattern is present in eukaryotes is unknown, largely due to the paucity of studies quantifying both genome-wide diversity and spontaneous mutation rates under natural conditions in organisms with different life histories and reproductive strategies." Its unclear from what is written why natural conditions nor life history are relevant here - can the authors provide citations and logic to back this?

We now revised the whole paragraph. We removed the information concerning life history and reproductive strategies in this paragraph but added more explanations in the discussion section. See lines 48-58 and lines 149-157.

L86 "mis-labeling during long-term maintenance of the duckweed collections" - is this a problem - Does it change any of your results - especially diversity?

Although this may affect the conclusion on where the sample was originally collected, this does not affect our results, as none of our analysis is based on the information of where the samples were originally collected. The diversity within and between populations were analyzed based on the population structure that was derived from the genomic information (independent of their geographic origin).

L93 - "distance between variants when their correlation coefficient (r^2) = 0.2" where does this cutoff for estimating LD plateau come from? It seems arbitrary and very high

We noticed an error in our original report. The cutoff value should be $r^2 = 0.33$ (1/3). This was suggested by Ardlie et al (2002) as a cut-off value, because r^2 -values above 1/3 indicate sufficiently strong LD to be useful for mapping. We now corrected this error and adjusted the reported LD values. See lines 93-96 and lines 258-263.

L103 - "Mutation rates can be markedly affected by outdoor environmental stresses such as temperature fluctuations and ultraviolet (UV) light 16-20" — These are strange references - as the UV experiments are artificial (not outdoor conditions) and the study on Epistasis is theory about selection and not relating to spontaneous mutation rates. - the premise that duckweed is stressed by environment and UV is a bit of a straw man argument. An organisms outdoor environment is what its been adapting too and there is no evidence provided it is "stressed" - I don't think you can say that mutation rate is likely lower in MA experiments. For many organisms not optimized for growth in the lab the conditions may be more stressful, without evidence - none of which is provided in this paper - can we say this is true.

To avoid the confusion, we now rephrased the sentences as: ". Because environmental factors, such as ultraviolet (UV) light, which prevails in the native habitats of *S. polyrhiza*, can affect mutation rates¹⁶⁻²⁰, mutation rates measured in the lab may not necessarily reflect mutation rate in the field. In an attempt to get an estimate of mutation rate more similar to what would be observed in the wild we estimated the genomic mutation rate in indoor and outdoor mutation accumulation (MA) experiments, and manipulated UV light in the outdoor experiments to further assess the effect of environmental factors (Figure S1 and External Dataset 2)." See lines 105-111.

L481 - "referes" – typo

Corrected.

It is unclear from the experimental description how the selection of fronds occurred between generations - A single frond was taken to start a new colony - but how was that frond selected - there is a chance for selection to enter the experiment if fronds with low growth rate are ignored, or those that accumulate mutations are less likely to contribute to the next generation. If the production of fronds is not synchronized how confident are the authors about the number of generations per transfer

We used an unbiased approach to select fronds during the MA experiments. Under the experimental conditions we used, duckweeds sequentially form new fronds approximately every three days. For all treatments, the frond that started to emerge first from the mother frond was propagated regardless of its own growth rate. Thus, the lineages were propagated in the absence of selection, except for mutations that are lethal in the recessive state, which should be extremely rare. We now added more details of the experimental procedures in the Method section. See lines 185-187.

L221 - "220. "MQ0 >= 4 && ((MQ0 / (1.0 * DP)) > 0.1) & QUAL < 30.0 & QD < 5.0"" - Can this be unpacked into normal words.

We now added the explanations for this. See lines 232-236.

L229 Expand on - "Loci with missing data" how many missing sites were required before the site was removed - how many sites did this affect? How many invariant sites are affected?

We removed any loci that had a single missing genotype (was missing in a single individual) across all populations. We removed 61281 SNPs, which is ~5% of SNPs for the analysis of population genetic statistics. The total number of invariant sites affected by other filtering was ~12 % of the genome size. We now adjusted all of the estimated Pi based on the callable sites. We also included this information in the method section. See lines 251-254.

L233 - For structure were loci sub-sampled? Linked sites should not be used to infer structure

We repeated our analysis with subsampled loci, in which we removed all loci that were linked ($r^2 > 0.33$). The main conclusion is the same – the most likely number of populations (K) was 4 and population assignment was the same as before, but the exact likelihood values for the structure plot changed slightly. We now included this in the method section and updated the structure plot in Figure 1. See lines 247-249 and Figure 1.

Throughout "Intra-specific" - is one word no hyphen, intraspecific

We now corrected this.

"Related individuals were removed and only SNPs with minor allele frequency (MAF) greater than 0.05 were kept" - does this apply only to the plink LD analysis

Yes, this was only applied to the plink LD analysis. We now clarified this point. See lines 256-258.

L241 - "The extent of useful LD for mapping can be defined as $r^2 = 0.240$ " - where does this statement come from?

As we explained above - The cutoff value should be $r^2 = 0.33$ (1/3). This was suggested by Ardlie et al (2002) as a cut-off value, because r^2 -values above 1/3 indicate sufficiently strong LD to be useful for mapping. We now corrected this error and adjusted the reported LD values. See lines 258-263.

L136 - "This relatively large N_e may have contributed to the evolution of a low mutation rate in *S. polyrhiza*, as selection can effectively drive down the mutation rate in populations with large N_e " The N_e is indeed large for a eukaryote but not for a unicellular organism - many of which have higher mutation rates despite having genomes dense with functional sequence and also high diversity. This relates to my "major" point above.

When the mutation rate (in coding regions) and effective population size are plotted among all organisms, there is a clear negative correlation. However, as we explained above, the efficacy of selection in reducing mutation rate, which is affected by recombination rate, background selection etc., can vary among species. Therefore, although overall mutation rates decrease with N_e among species, variations do exist.

143-148 I think the author's thinking is backwards - if recessive mutations are sheltered, how can they drive the mutation rate to evolve lower? I would expect less selection on a mutator allele.

We now revised the referred sentence. See lines 149-157

L255 " U^n where U is the mutation rate, and n refers to number of samples" I think this is not the correct equation. There are multiple generations and you only need two samples to have the mutation, not all n

This was due to a wording error. Here, n refers to number of samples that have the same mutation, but not the total sample size. We now corrected this error in the text. See lines 276-277.

References:

1. Waples RS, Do C. Linkage disequilibrium estimates of contemporary N_e using highly variable genetic markers: a largely untapped resource for applied conservation and evolution. *Evol Appl* **3**, 244-262 (2010).
2. Siol M, *et al.* Patterns of Genetic Structure and Linkage Disequilibrium in a Large Collection of Pea Germplasm. *G3 (Bethesda)* **7**, 2461-2471 (2017).
3. Yan J, Shah T, Warburton ML, Buckler ES, McMullen MD, Crouch J. Genetic characterization and linkage disequilibrium estimation of a global maize collection using SNP markers. *PLoS One* **4**, e8451 (2009).
4. Ewing AD, *et al.* Combining tumor genome simulation with crowdsourcing to benchmark somatic single-nucleotide-variant detection. *Nat Methods* **12**, 623-630 (2015).
5. Stocks M, Siol M, Lascoux M, De Mita S. Amount of information needed for model choice in Approximate Bayesian Computation. *PLoS One* **9**, e99581 (2014).
6. Nelson CW, Moncla LH, Hughes AL. SNPGenie: estimating evolutionary parameters to detect natural selection using pooled next-generation sequencing data. *Bioinformatics* **31**, 3709-3711 (2015).

Reviewers' Comments:

Reviewer #1:

Remarks to the Author:

I have read over the authors' replies to my comments, and the comments of the other reviewers. The authors have satisfactorily addressed my concerns, with one minor exception. The new method for estimating the false negative rate is a good idea. However, since 51 of the 1000 introduced mutations "resulted in low coverage and were dropped", it seems like the false negatives should include the sites that were dropped. I think that would increase the false neg rate from 2.7% to 7.7%. Which is still too low to change the basic conclusions. I may be wrong about this, but I'd like to hear the explanation of why the dropped sites don't count as false negatives. With respect to my earlier comment about the 20% of the genome not covered, if mutations are more likely to occur in regions that are poorly covered, that has implications for the overall estimated rate.

Reviewer #2:

Remarks to the Author:

Thanks to the authors for their detailed responses to the reviewers' comments. I am satisfied with the revision.

Reviewer #3:

Remarks to the Author:

I have re-reviewed the paper and I am impressed with the efforts made by the authors to address the criticisms. I think most of what I would want altered relates to the main interpretation of the results. I am still not convinced by the DBH explaining the data. However, that said, I think it is a possible explanation, so rather than not publishing the paper it would be good to perhaps explore and contrast their results with the alternate hypothesis I reference below. The community can decide what they think. Below I outline my response to the three criticisms I originally outlined

1. N_e and μ

My concern mostly remains over the explanation that N_e is so high that μ has been driven so low that diversity is reduced. If the CI for the correlation coefficient beta is between -0.6 and 0.89 overall and -0.06 and -1.15 for multicellular eukaryotes, it seems a stretch to consider it an obvious explanation that duckweed is the first species with such a strong negative correlation. Moreover, most arguments one can make about its mating system and genome structure equally apply to *Chlamydomonas* which has a more compact genome, higher mutation rate and 2 orders of magnitude higher N_e . - The authors are looking at the extreme end of a confidence interval that spans from, no effect of N_e on μ to such a strong effect of N_e on μ that they can explain their results. In order for the relationship between N_e and μ to be as strong as they suggest this species of duckweed would need to have very deleterious mutations - in fact probably the strongest selection against mutation of any species described.

The alternate explanation, that I think is more parsimonious, is that the low diversity is caused by selection at linked sites. Given the strong selection that the authors argue and the low frequency of sex, selection at linked sites could be reducing diversity genome wide. This explanation for 'Lewontin's paradox' has been argued in the literature (e.g. Natural Selection Constrains Neutral Diversity across A Wide Range of Species Corbett-Detig et al 2015 PLoS Genetics" and I think applies here better than assuming that duckweed has an unusual distribution of fitness effects (see Extreme Lewontin's

Paradox in Ubiquitous Marine Phytoplankton Species" Dmitry A Filatov MBE).

2. False Negative Rate:

For the false negative rate - presumably real mutations are always seen as heterozygotes in MA - but the distribution of ancestral and mutated allele frequencies used in the simulations is not reported. In fact it is not reported whether the mutations were introduced as heterozygotes at all. If they are introduced as homozygous then the test is not valid and if they are introduced at exactly 50:50 or based on the highest quality heterozygotes than its biased towards achieving heterozygous calls.

3. Equilibrium:

While I agree the overall correlation between μ and N_e remains despite deviations from equilibrium there is noise around the line, if you examine one species, π/μ may not reflect N_e at all - that said, I am content with the investigations into the SFS as decent evidence for a population not terribly out of equilibrium - I was mostly concerned that the duckweed population may have expanded greatly and the SFS would have many rare variants demonstrating that diversity doesn't reflect $4N_e\mu$

Reviewers' comments:

Reviewer #1 (Remarks to the Author):

I have read over the authors' replies to my comments, and the comments of the other reviewers. The authors have satisfactorily addressed my concerns, with one minor exception. The new method for estimating the false negative rate is a good idea. However, since 51 of the 1000 introduced mutations "resulted in low coverage and were dropped", it seems like the false negatives should include the sites that were dropped. I think that would increase the false neg rate from 2.7% to 7.7%. Which is still too low to change the basic conclusions. I may be wrong about this, but I'd like to hear the explanation of why the dropped sites don't count as false negatives. With respect to my earlier comment about the 20% of the genome not covered, if mutations are more likely to occur in regions that are poorly covered, that has implications for the overall estimated rate.

Thanks for your constructive suggestions.

As our original goal was to evaluate the false negatives rate from our variant filtering process, we did not consider the "dropped loci" from our original false negative analysis. These loci were not spiked into the BAM files by BAMsurgeon due to insufficient input / output read coverage. This was due several different reasons. One was due to a few number of single reads (0.5%) that remained in the BAM file (BAMsurgeon cannot handle mixed single and paired-end reads), and the second one was due to non-unique mapping.

After investigating the detailed information of the dropped loci, we noticed that while the loci that were dropped due to the first reason can be ignored from the false negative analysis, the loci that were dropped due to the second reason should be included for calculating the false negative rate.

Therefore, we repeated the simulation with a modified mutation spiking parameter (in BAMsurgeon, used --force option) and re-estimated the false negative rate. Among 1000 input loci, we could identify 945 of them, which gives false negative rate of 5.5% (95% CI: 4.1-7.2%). We now updated the false negative estimation in the revision.

In addition, we also incorporated the suggestions from reviewer 3 and varied the allele frequencies of the alternative alleles.

Concerning the 20% of genome that was not covered, we agree, if the mutations are more likely to occur in these uncovered regions, it might indeed affect the overall estimated rate. However, these regions contain mostly repetitive sequences and similar to other studies on mutation rate we cannot accurately estimate rate of mutation in regions that are difficult to sequence and assemble.

In addition, from a comparative point of view, the total repetitive sequences in *S. polyrhiza* is among the lowest of all sequenced multi-cellular eukaryotes. If the repetitive sequences have higher mutation rate in general, it will result in an underestimation of the mutation rates in all organisms, but the effect is smallest in *S. polyrhiza*. Therefore, we believe, even if the repetitive regions have a higher mutation rate, the main conclusions that we draw in our manuscript will likely still hold true.

We now updated the false negative estimation in the revision. See texts highlighted in red.

Reviewer #2 (Remarks to the Author):

Thanks to the authors for their detailed responses to the reviewers' comments. I am satisfied

with the revision.

Thanks for this positive feedback.

Reviewer #3 (Remarks to the Author):

I have re-reviewed the paper and I am impressed with the efforts made by the authors to address the criticisms. I think most of what I would want altered relates to the main interpretation of the results. I am still not convinced by the DBH explaining the data. However, that said, I think it is a possible explanation, so rather than not publishing the paper it would be good to perhaps explore and contrast their results with the alternate hypothesis I reference below. The community can decide what they think. Below I outline my response to the three criticisms I originally outlined

1. Ne and μ

My concern mostly remains over the explanation that Ne is so high that μ has been driven so low that diversity is reduced. If the CI for the correlation coefficient beta is between -0.6 and 0.89 overall and -0.06 and -1.15 for multicellular eukaryotes, it seems a stretch to consider it an obvious explanation that duckweed is the first species with such a strong negative correlation. Moreover, most arguments one can make about its mating system and genome structure equally apply to Chlamydomonas which has a more compact genome, higher mutation rate and 2 orders of magnitude higher Ne. - The authors are looking at the extreme end of a confidence interval that spans from, no effect of Ne on μ to such a strong effect of Ne on μ that they can explain their results. In order for the relationship between Ne and μ to be as strong as they suggest this species of duckweed would need to have very deleterious mutations - in fact probably the strongest selection against mutation of any species described.

The alternate explanation, that I think is more parsimonious, is that the low diversity is caused by selection at linked sites. Given the strong selection that the authors argue and the low frequency of sex, selection at linked sites could be reducing diversity genome wide. This explanation for 'Lewontin's paradox' has been argued in the literature (e.g. Natural Selection Constrains Neutral Diversity across A Wide Range of Species Corbett-Detig et al 2015 PLoS Genetics" and I think applies here better than assuming that duckweed has an unusual distribution of fitness effects (see Extreme Lewontin's Paradox in Ubiquitous Marine Phytoplankton Species" Dmitry A Filatov MBE).

We agree that the linked selection could be an alternative/additional reason that have resulted in the overall low genetic diversity. We now added this in our discussion.

2. False Negative Rate:

For the false negative rate - presumably real mutations are always seen as heterozygotes in MA - but the distribution of ancestral and mutated allele frequencies used in the simulations is not reported. In fact it is not reported whether the mutations were introduced as heterozygotes at all. If they are introduced as homozygous then the test is not valid and if they are introduced at exactly 50:50 or based on the highest quality heterozygotes than its biased towards achieving heterozygous calls.

Thanks for raising this point. We missed this important information in our previous revision. Indeed, all simulated mutations were introduced as heterozygous, with average frequency of alternative alleles = 0.5. Now, we performed an additional round of simulation by considering the variations of alternative allele frequencies (standard deviation = 0.1). The new simulation showed that the false negative rate is 5.5%, slightly higher than the previous estimate. In the latest revision, we updated the results from the latest simulation.

3. Equilibrium:

While I agree the overall correlation between μ and N_e remains despite deviations from equilibrium there is noise around the line, if you examine one species, π/μ may not reflect N_e at all - that said, I am content with the investigations into the SFS as decent evidence for a population not terribly out of equilibrium - I was mostly concerned that the duckweed population may have expanded greatly and the SFS would have many rare variants demonstrating that diversity doesn't reflect $4N_e\mu$

We calculated the SFS and estimated the demographic changes of the four *S. polyrhiza* populations using Stairwayplot 2. The results suggest that, most of the populations experienced a moderate level of population expansions/contractions (See figure below). The results are consistent with the Tajima's D estimations. However, as both relatively small sample size of each population and migrations among populations might affect the predicted demographic changes, we are reluctant to include these results in our current manuscript without further supporting evidence.

Figure S1. Demographic models of the four giant duckweed populations. Time and estimated effective population size are shown on x-axis and y-axis, respectively. Lines indicate 5% and 25% confidence intervals from 200 bootstraps. The N_e was estimated using Stairway plot2 (<https://sites.google.com/site/jpopgen/stairway-plot>) based on the unfolded site frequency spectrum (SFS) of each population. The SFSs were calculated from easySFS (<https://github.com/isaacovercast/easySFS>).

Reviewers' Comments:

Reviewer #1:

Remarks to the Author:

I have read the authors' replies to my comments and to those of REviewer #3 and I am satisfied that the paper is suitable for publication.

Reviewer #3:

Remarks to the Author:

Ne and μ

I think it is good that the authors have added a new paragraph about selection at linked sites, but the paragraph is not well written. There are quite a few corrections now needed.

There is no mention of the widespread effect of background selection which is important in explaining genome wide patterns. There are also no references which is poor scholarship.

The sentence "An indication of strong linked selection in *S.polyrhiza* is the relatively high genome-wide π_N / π_S ratio" doesn't really make sense to me. Strong background selection is a result of strong purifying selection but the high π_N / π_S ratio is not indicative of purifying selection - which implies low levels of purifying selection and little background selection. On the other hand, strong purifying selection on synonymous sites might drive up the ratio which is plausible in a species with $N_e = 1e6$. Moreover, selective sweeps reduce π_S but $\pi_N : \pi_S$ ratio shouldn't really be effected - so I don't follow the argument. Lastly, the high $\pi_N : \pi_S$ doesn't line up with previous arguments that the mutation rate is driven to very low levels because mutations are more strongly deleterious in this species.

L168 "The effect" not "the effects"

169" However, the effects of linked selection on reducing neutral diversity is determined by the frequency and trajectory of selective sweeps" This isn't true - you haven't included background selection. Also the effects of linked selection are determined by the frequency of sweeps, the recombination rate, the strength of selection and the age of the beneficial allele - I don't know what is meant by trajectory.

L171 "long-term" not "long terms"

L172 "extend to what" should be "extent to which"

REVIEWERS' COMMENTS:

Reviewer #1 (Remarks to the Author):

I have read the authors' replies to my comments and to those of REviewer #3 and I am satisfied that the paper is suitable for publication.

Thanks.

Reviewer #3 (Remarks to the Author):

Ne and μ

I think it is good that the authors have added a new paragraph about selection at linked sites, but the paragraph is not well written. There are quite a few corrections now needed.

There is no mention of the widespread effect of background selection which is important in explaining genome wide patterns. There are also no references which is poor scholarship.

We now added the missing references.

The sentence "An indication of strong linked selection in *S. polyrhiza* is the relatively high genome-wide $\pi N / \pi S$ ratio" doesn't really make sense to me. Strong background selection is a result of strong purifying selection but the high $P_n P_s$ ratio is not indicative of purifying selection - which implies low levels of purifying selection and little background selection. On the other hand, strong purifying selection on synonymous sites might drive up the ratio which is plausible in a species with $N_e = 1e6$. Moreover, selective sweeps reduce P_s but $P_n : P_s$ ratio shouldn't really be effected - so I don't follow the argument. Lastly, the high $pN : pS$ doesn't line up with previous arguments that the mutation rate is driven to very low levels because mutations are more strongly deleterious in this species. We appreciate this critical comment. The confusion was due to unclear wordings. Linked selection, either genetic hitchhiking or background selection, can reduce genetic diversity. However, the effects on deleterious variants and neutral variants can be different. Because deleterious variants segregate at lower frequencies than neutral variants in populations, after recurrent linked selection events (either genetic hitchhiking or background selection), deleterious variants are expected to reach their equilibrium relatively faster than neutral variants. Proportionally, linked selection will result in a higher loss of diversity for neutral variants than deleterious variants. As a consequence, the ratio of $\pi N / \pi S$ will be higher. However, several other factors may also contribute to the changes of $\pi N / \pi S$, such as codon usage bias, selection on synonymous sites etc., and our current data is not sufficient to estimate the factors that determine the linked selection (both genetic hitchhiking and background selection). Currently, we are working on a large whole-genome resequencing dataset of *S. polyrhiza*, which might allow us to address these questions more confidently in near future. However, they are beyond the scope of this manuscript.

We now removed the questioned sentence in the manuscript and revised the paragraph accordingly.

L168 "The effect" not "the effects"

Corrected.

169" However, the effects of linked selection on reducing neutral diversity is determined by the frequency and trajectory of selective sweeps" This isn't true - you haven't included background selection. Also the effects of linked selection are determined by the frequency of sweeps, the recombination rate, the strength of selection and the age of the beneficial allele - I don't know what is meant by trajectory.

We now specifically mentioned background selection and changed the wordings.

L171 "long-term" not "long terms"

Corrected.

L172 "extend to what" should be "extent to which"

Corrected.